# Magnetically Stimulated Myogenesis Recruits a CRY2-TRPC1 Photosensitive Signaling Axis

**DOI:** 10.3390/cells14030231

**Published:** 2025-02-06

**Authors:** Jan Nikolas Iversen, Yee Kit Tai, Kwan Yu Wu, Craig Jun Kit Wong, Hao Yang Lim, Alfredo Franco-Obregón

**Affiliations:** 1Department of Surgery, Yong Loo Lin School of Medicine, National University of Singapore, Singapore 119228, Singapore; nikolas.iversen@u.nus.edu (J.N.I.); lesleywu@nus.edu.sg (K.Y.W.);; 2Institute of Health Technology and Innovation (iHealthtech), National University of Singapore, Singapore 117599, Singapore; 3Biolonic Currents Electromagnetic Pulsing Systems Laboratory (BICEPS), National University of Singapore, Singapore 117599, Singapore; limh7@student.unimelb.edu.au; 4NUS Centre for Cancer Research, Yong Loo Lin School of Medicine, National University of Singapore, Singapore 117599, Singapore; 5Department of Biochemistry and Pharmacology, School of Biomedical Sciences, University of Melbourne, Melbourne, VIC 3010, Australia; 6Department of Physiology, Yong Loo Lin School of Medicine, National University of Singapore, Singapore 117593, Singapore; 7Competence Center for Applied Biotechnology and Molecular Medicine, University of Zürich, 8057 Zürich, Switzerland

**Keywords:** cryptochromes, magnetoreception, circadian rhythm, pulsed electromagnetic fields, cell development, radical pair mechanism, photomodulation, myogenesis, melanopsin, proliferation

## Abstract

The cryptochromes are flavoproteins that either individually or synergistically respond to light and magnetic field directionality as well as are implicated in circadian rhythm entrainment and development. Single brief exposures (10 min) to low energy (1.5 mT) pulsed electromagnetic fields (PEMFs) were previously shown to enhance myogenesis by stimulating transient receptor potential canonical 1 (TRPC1)-mediated Ca^2+^ entry, whereby downwardly directed fields produced greater myogenic enhancement than upwardly directed fields. Here, we show that growth in the dark results in myoblasts losing their sensitivity to both magnetic field exposure and directionality. By contrast, overexpressing or silencing cryptochrome circadian regulator 2 (CRY2) in myoblasts enhances or reduces PEMF responses, respectively, under conditions of ambient light. Reducing cellular flavin adenine dinucleotide (FAD) content by silencing riboflavin kinase (RFK) attenuated responsiveness to PEMFs and inhibited selectivity for magnetic field direction. The upregulation of TRPC1 and cell cycle regulatory proteins typically observed in response to PEMF exposure was instead attenuated by upwardly directed magnetic fields, growth in the darkness, magnetic shielding, or the silencing of CRY2 or RFK. A physical interaction between CRY2 and TRPC1 was detected using coimmunoprecipitation and immunofluorescence, revealing their co-translocation into the nucleus after PEMF exposure. These results implicate CRY2 in an identified TRPC1-dependent magnetotransduction myogenic cascade.

## 1. Introduction

The cryptochromes are a family of flavoproteins expressed in plants and animals that have been implicated in the detection of light and magnetic field anisotropy as well as in circadian rhythm entrainment and development [1]. Nevertheless, “how?” and “in what context do the cryptochromes consolidate such apparently disparate roles?” remained largely “cryptic”. With reference to magnetic sensitivity in the context of light, Ahmad and colleagues [2] elegantly showed that wild-type *Drosophila* larvae avoided a region of a dish that was continually exposed to 1.8 milliTesla (mT) pulsed electromagnetic fields (PEMFs) when submitted to blue light (<500 nm), but not red light (>600 nm), an effect that was precluded by shielding the larvae from the delivered PEMFs. On the other hand, mutant fruit fly larvae deficient in cryptochromes did not exhibit this avoidance behavior to PEMFs, whereas cryptochrome-deficient transgenic fly larvae heterologously expressing human cryptochromes regained the avoidance response. Therefore, both insect and mammalian cryptochromes were capable of conferring some level of magnetic sensitivity, manifested as an avoidance response, in the presence of blue light.

With reference to light and in the context of myogenesis, it was also previously shown that 435 nm light could provoke contractures of myotubes derived from the iris (eye pupillary sphincter) or pectoral striated muscles [3]. This photomechanical transduction response required the expression of the mammalian cryptochromes 1 and 2, CRY1 and CRY2, as well as the presence of reduced flavins, and was dependent on extracellular Ca^2+^ entry and release from intracellular stores. CRY2 has also been shown to be required for light- and Ca^2+^-sensitive circadian rhythm-regulated muscle development that further implicated the involvement of a cation channel in the response cascade [4]. Accordingly, photomodulated (420 nm) muscle development requires the involvement of calcium-permeable transient receptor potential (TRP) channels, in particular TRPC1 [5], an intracellular store-operated Ca^2+^ channel [6] involved in cellular mechanotransduction [7,8] and magnetoreception [9]. Provocatively, TRPC1 has been implicated in the mechanical photoresponses of vertebrate rod cells that result in the shortening of the photoreceptor outer segment in response to light or mechanically invoked Ca^2+^ entry [10]. Lastly, performing a pathway enrichment analysis across multiple tissues, Birnbaumer and colleagues [11] revealed an association between the expressions of TRPC family members and components of the circadian rhythm pathway, including CRY2. In sum, these diverse data reveal a complex interplay between diverse biophysical stimuli, such as light, magnetic fields, and mechanical forces, in the elaboration of CRY-mediated developmental responses in skeletal muscle. What remained to be resolved was the nature of the transduction step converting photoactivation of the cryptochromes into myogenic responses.

Brief exposure to PEMFs (1.5 mT for 10 min) was previously shown to promote in vitro myogenesis, an effect that was mediated by the opening of TRPC1 channels [9]. Pharmacological antagonism of TRPC1 or genetic silencing of TRPC1, but not TRPM7, precluded the induction of myogenesis by magnetic fields [9]. Moreover, the TRPC1 expression pattern coincided with responsiveness to magnetic fields under numerous developmental scenarios. Shielding myoblasts from all ambient magnetic fields reduced basal TRPC1 expression, whereas a single brief PEMF exposure augmented TRPC1 expression relative to unshielded non-exposed cells. A vital component of the magnetic mitohormetic response was a preference for magnetic field directionality, exhibiting the greatest responses to magnetic field lines flowing in the downward direction [12]. TRPC1 expression was most strongly augmented in response to downward magnetic field exposure, compared to upwardly or horizontally directed exposure, which further coincided with the greatest myogenic enhancement [12]. This magnetic myogenic response invoked a TRPC1-mitochondrial signaling cascade that instilled mitohormetic adaptations governing cell survival and development. Accordingly, targeted vesicular delivery of TRPC1 reinstated the lost magnetic induction of mitochondrial respiration and myoblast proliferation demonstrated in CRISPR-Cas9 TRPC1 knockdown myoblast cell lines [13]. Taken together, these diverse lines of evidence indicate that TRPC1 expression is necessary and sufficient to confer magnetic sensitivity. Finally, synergism between magnetic field exposure and light has also been previously demonstrated that shared a reliance on TRPC1 function and expression [14]. Available evidence is hence consistent with functional synergisms between cryptochrome magnetically tuned photoactivation and TRPC1-mediated biophysical signal transduction.

The objective of this study was to investigate the potential contribution of CRY2 in an established magnetoreceptive developmental cascade requiring the participation of TRPC1. To this end, we examined the TRPC1-mediated myogenic responses after shielding myoblasts from both light and magnetic fields or by altering the genetic expression of CRY2 or riboflavin kinase (RFK) responsible for the synthesis of FAD, an essential cofactor for CRY2 function. Changes in the magnetic field directionality selectivity were specifically noted. Finally, we looked for evidence of physical interaction between TRPC1 and CRY2 that would provide information as to their mode of interaction to assist in this magnetically tuned myogenic response.

## 2. Materials and Methods

### 2.1. Cell Culture and Pharmacology

C2C12 mouse skeletal myoblasts (American Type Culture Collection; LGC Standards, Teddington, UK) were grown and maintained as previously described [9]. Unless otherwise stated, myoblasts were seeded at a density of 3000 cells/cm^2^ and exposed to PEMFs 24 h after plating at a cell confluency below 40%. Importantly, as the aminoglycoside antibiotics, like streptomycin, have been shown to inhibit TRPC channels [15], which are necessary for magnetic induction [9], they were omitted from the culture medium.

### 2.2. Cell Count

Cell enumeration was performed using a trypan blue exclusion assay as previously described [9,12]. Cell counting was performed 24 h following magnetic exposure and is represented as technical replicates of three per indicated number of biological replicates in each designated figure legend unless otherwise stated.

### 2.3. Generation of Plasmid and Stable Cell Line

The full-length CRY2 cDNA (Accession: AF156987.1; 1770 base pairs) was amplified by PCR to create the GFP-CRY2 and CRY2-FLAG plasmids. SacI and SalI restriction enzymes were used to subclone the amplified cDNA into the pEGFP-C1 vector directionally, and XhoI and SalI restriction enzymes were used to subclone a FLAG plasmid with a pEGFP-N1 backbone (with the GFP sequence removed and replaced with a FLAG sequence). Lipofectamine 3000 reagent (Invitrogen, Thermo Fisher Scientific, Waltham, MA, USA) was used to transfect C2C12 cells with the plasmids. For the generation of stable cell lines, GFP and GFP-CRY2 transfected cells were selected 48 h after transfection with 1 mg/mL Geneticin (Gibco, Thermo Fisher Scientific, Waltham, MA, USA). GFP-positive cells were enriched using a cell sorter (Moflo Astrios; Beckman Coulter, Brea, CA, USA). Stable cells were maintained in selection media containing 1 mg/mL Geneticin for at least 2 weeks before characterization of the cells. The overexpression of GFP-CRY2 in the stable cells was characterized using Western blot and qPCR analyses, illustrated in Appendix A.

### 2.4. PEMF Exposure Paradigms

The experimental basis for the magnetic signal parameters, exposure duration, directionality, and timing of the PEMF exposure paradigm used in the present manuscript was previously described [9,12,16]. The PEMF device and coil orientation are illustrated in Appendix A. Unless otherwise stated, all samples were exposed to PEMFs for 10 min at room temperature. Time-matched control samples (0 mT) were placed within the PEMF device when not activated [9] and were otherwise handled identically. The coil system, when in operation, does not generate heat or mechanical vibrations [9].

### 2.5. Light, Dark, and µ-Box Apparatus and Paradigms

Cells were seeded into lighted, darkened, or magnetically shielded conditions using transparent, black, or µ-metal boxes, respectively. The µ-metal box was previously described [9]. The black and transparent boxes were designed to match the µ-metal box, with modifications to control light exposure. The black box had gas exchange portals positioned under the lid’s overhanging lip to minimize light entry (Appendix A). The clear box was identical to the black box but allowed light to pass through. Both boxes were fabricated by 3D Print Singapore LLP using PETG (FDM technology) for the black box and Somos WaterShed XC 11,122 (SLA technology) for the transparent box. The boxes measured 15.6 cm (length) by 13.2 cm (width) by 5 cm (height) with covers measuring 16 cm (length) by 13.6 cm (width) by 1.3 cm (height). Ambient laboratory lighting originated from a fluorescent light tube (Lifemax TL-D 36W/54-765 1SL/25; Philips, Amsterdam, The Netherlands). The light tube color temperature is 6500K to produce a cool daylight setting, where the distribution of wavelengths is 450–495 nm for blue light, 495–570 nm for green light, and 570–700 nm for yellow and red light. The peak wavelength at 6500K is in the blue to green light range at 480–490 nm, the blue component being the most dominant.

### 2.6. Western Analysis

Cell lysates were prepared in ice-cold RIPA buffer supplemented with phosphatase inhibitors as previously described [14]. The protein concentration of the soluble fractions was determined by BCA reagent (Thermo Fisher Scientific, Waltham, MA, USA). A total of 25–50 µg of total protein was resolved using 10–12% SDS-PAGE and transferred to PVDF membranes (Bio-Rad, Hercules, CA, USA). Membranes were blocked using 5% low-fat milk or BSA in TBST containing 0.1% Tween-20. Primary antibodies were diluted in SuperBlock TBS (Thermo Fisher Scientific) as listed in Table 1. HRP-conjugated anti-rabbit or anti-mouse secondary antibodies were diluted (1:3000, Thermo Fisher Scientific) in 5% milk. Membranes incubated with SuperSignal West ECL or West Femto chemiluminescent substrate (Thermo Fisher Scientific) were analyzed using LI-COR Image Studio.

### 2.7. Real-Time qPCR

Quantitative reverse–transcription polymerase chain reaction (RT-qPCR) and mRNA reverse transcription were carried out as previously described [9]. Transcript expression was quantified using SSoAdvanced Universal SYBR Green (Bio-Rad) on a CFX Touch Real-Time PCR Detection System (Bio-Rad). Relative transcript expression was determined using the 2^−ΔΔCt^ method, with β-2 microglobulin (B2M) as the reference housekeeping gene. The qPCR primers used are listed in Table 2.

### 2.8. CRY2 and RFK Silencing

The silencing of the *CRY2* and *RFK* genes in C2C12 cells was performed using three pre-designed dicer–substrate short interfering RNAs (dsiRNA; IDT, Newark, NJ, USA). Each dsiRNA targeted the coding sequence of *CRY2* (NC_000068.8) and *RFK* (NM_019437.3). Transfection was performed using Lipofectamine 3000 reagent (Invitrogen) as per the manufacturer’s protocol. *CRY2*- and *RFK*-silenced cells were validated using qPCR. qPCR quantification showed that one out of three pre-designed dsiRNAs was able to knock down ~50% of the *CRY2* gene and ~95% of the *RFK* gene as compared to the scrambled/negative control and was thus used for this protocol (Appendix A).

### 2.9. Immunofluorescence Using Laser Confocal Imaging

Myoblasts were seeded onto coverslips at a density of 70,000 cells/well 24 h prior to PEMF or sham exposure for 10 min. At designated timepoints, cells were fixed in 4% paraformaldehyde for 15 min and then permeabilized with 0.1% Triton in PBS for 15 min and blocked in SuperBlock TBS (Thermo Fisher Scientific). CRY2 (1:50; Proteintech, Rosemont, IL, USA) and TRPC1 (1:50; Santa Cruz, Dallas, TX, USA) antibodies were incubated overnight and subsequently treated with Alexa Fluor 488 and 594 secondary antibodies, respectively (1:500; Thermo Fisher Scientific). Nuclei of cells were co-stained with DAPI (Sigma-Aldrich, St. Louis, MO, USA). Cells were finally mounted and visualized using a laser scanning confocal microscope (Olympus, Tokyo, Japan). CRY2 expression was quantified by determining the mean signal intensity per cell normalized to the number of nuclei for a minimum of 10 cells per view across 3 or more technical replicates.

### 2.10. Subcellular Fractionation and Immunoprecipitation Assay

C2C12 cells were grown in 100 mm culture plates until 80% confluency. Cells were rinsed with ice-cold PBS before the addition of 500 µL of subcellular fractionation (SF) buffer, containing 250 mM sucrose, 20 mM HEPES (pH 7.4), 10 mM KCl, 1.5 mM MgCl_2_, 1 mM EDTA, 1 mM EGTA, 1 mM DTT, and cocktail protease inhibitor (Nacalai Tesque, Kyoto, Japan). Cell lysates were agitated at 4 °C for 30 min using a tube roller (50 rpm) and were subsequently centrifuged at 720 g for 5 min. The supernatant was kept as the cytosolic and membrane fraction. The cell pellet was resuspended in 500 µL nuclear fractionation (NF) buffer containing 50 mM Tris HCl (pH 8), 150 mM NaCl, 1 mM EDTA, 1% IGEPAL CA-630, 5% glycerol, and cocktail protease inhibitor. The resuspended pellet was sonicated on ice using a 3-second on/off protocol for a total of 3 cycles, 30% amplitude on the sonic dismembrator (Thermo Fisher Scientific). Fractionated lysates were further diluted in immunoprecipitation (IP) buffer containing 25 mM Tris HCl, pH 8, 150 mM NaCl, 1% Triton X-100, and 5% glycerol. To pre-clear the lysates, 25 µL of protein A/G PLUS Agarose (Santa Cruz) was added per 1 mL of resulting lysates and left to rotate for 1 h at 4 °C. After discarding the beads, the supernatant was centrifuged for 5 min at 2500 rpm and divided equally for pulldown using FLAG (1:50; Proteintech) antibody overnight in 4 °C. Furthermore, 25 µL of protein A/G PLUS Agarose was added. After 2 h of agitation, the tubes were centrifuged for 1 min at 3000 rpm. The pelleted beads were rinsed several times in PBS containing 0.1% Tween-20 and 5% glycerol. The beads were reconstituted in 40 µL of 1X loading buffer before resolving them using SDS-PAGE gel. Clean-Blot IP Detection Reagent (Thermo Fisher Scientific) at a dilution of 1:2000 was used as a secondary antibody.

### 2.11. Statistical Analysis

All experiments were performed in biological triplicates per condition, unless otherwise stated. Statistical analyses were carried out using GraphPad Prism (Version 10.2.0 for Windows, GraphPad Software). Data normality was assessed using the Shapiro–Wilk test, and all data were assumed to follow a normal distribution. Unless otherwise stated, one-way analysis of variance (ANOVA) with Šidák’s post hoc test was used to compare the values across multiple groups. For the comparison between two independent samples, Student’s t-test was performed. Identification of outliers was performed using the ROUT method, and only obvious outliers were removed.

## 3. Results

The contribution of light, or lack thereof, over the usual proliferative enhancement observed in response to PEMF exposure [9] was examined with reference to magnetic field direction [12]. In agreement with previous studies [12], under conditions of normal laboratory lighting, downward magnetic field exposure produced the greatest enhancement in cell proliferation of 67%, whereas upward magnetic field exposure increased cell number by 40% relative to the basal proliferation of unexposed myoblasts (Figure 1). By contrast, PEMF exposure of myoblasts seeded and grown in darkness exhibited proliferative responses of 39% and 22% for magnetic fields in the downwards and upwards, respectively, that was accompanied by a reduction in sensitivity to field directionality.

We next examined the effects that growth in the dark exerted over proteins involved in the mammalian circadian response governed by the cryptochromes [4,17] and correlated these to changes in the expression of genes and proteins previously shown to be invoked by brief PEMF exposure under normal lighting conditions [9]. After 48 h of growth in the dark, cell proliferation was significantly decreased relative to myoblast cultures grown under ambient light conditions (Figure 2A). The protein levels of myogenic proliferation and differentiation regulators were also reduced by growth in the dark relative to growth in lighted conditions, including TRPC1, Cyclin B1, Cyclin D1, phosphorylated ERK, and MyoD protein (Figure 2B). Gene expression levels of the circadian rhythm regulators *CRY2*, *PER1*, *PER3, BCLAF*, *CLOCK,* and *TIMELESS* were also decreased in myoblasts grown in the dark (Figure 2C). Ambient lighting per se was hence permissive for PEMF responses.

The consequences of CRY2 overexpression were next examined. A C2C12 cell line overexpressing a GFP-CRY2 fusion protein was generated and compared against a stable cell line expressing a GFP vector alone (mock transfection). Under unstimulated conditions, the CRY2-overexpressing cells exhibited greater numbers than the mock-transfected cells, red hatched and red solid bars, respectively (Figure 3A). Upon PEMF exposure (blue), moreover, the CRY2 overexpressors showed a greater increase in cell number relative to unexposed cells (red) (Figure 3A). The CRY2 overexpressors (hatched) also showed greater increases in the protein levels of TRPC1, Cyclin B1, and phosphorylated ERK in response to PEMF exposure as well as an increase in Cyclin D1 levels in the unexposed state (Figure 3B). Phosphorylated ERK is produced by the Ras/ERK signaling pathway in response to proliferation induction, requiring the participation of the cell cycle regulators, Cyclin B1 and Cyclin D1 [18]. CRY2 overexpression thus increased the magnetic induction of TRPC1 and cell cycle regulatory proteins previously shown to be upregulated by the combination of PEMF and light exposures [4,14].

It was previously shown that growing myoblasts within a µ-metal container slowed their basal proliferation as well as precluded response to magnetic exposure due to the ability of the material to block the penetrance of both ambient and delivered magnetic fields [9]. C2C12-CRY2-overexpressing and mock-transfected cells were seeded into conventional tissue culture dishes and next immediately placed into a similarly dimensioned container that prevented the penetration of light (“dark”) (Appendix A). Under control conditions (0 mT), cell proliferation was greater in CRY2-overexpressing cells (hatched red bar) but was reduced by growth in the dark in both CRY2-overexpressing (hatched black bar) and mock-transfected (solid black bar) cells and by magnetic isolation in CRY2-overexpressing cells (hatched gray bar; µ-metal) (Figure 4). On the other hand, PEMF exposure increased the proliferation rate beyond all other unexposed conditions, particularly in response to CRY2 overexpression (hatched blue bar). Specifically, exposure to PEMFs increased the proliferation of GFP and GFP-CRY2 cells by 25% and 38%, respectively, compared to their respective unexposed scenarios, and by another 38% in GFP-CRY2 relative to GFP cells. GFP cells shielded from light or magnetic fields showed reductions in cell number of 10% and 6%, respectively, relative to their respective ambient scenario (solid red), and CRY2 overexpressors shielded from light or magnetic fields showed reductions in cell number of 18% and 13%, respectively, relative to their respective ambient scenario (hatched red). Changes in myogenic protein expression mirrored the trends observed for cell proliferation for TRPC1, Cyclin B1, Cyclin D1, and phosphorylated ERK (Figure 4B), whereby CRY2-overexpressed myoblasts displayed significant PEMF-induced increases in these proteins compared to the dark or magnetic field shielding conditions, and GFP cells failed to reach statistical significance. Overall, these results demonstrate that CRY2 overexpression enhanced magnetic responsiveness (TRPC1) and myogenic proliferation (Cyclin B1, Cyclin D1, and phosphorylated ERK). On another level, growth in the dark produced a stronger inhibition of cell proliferation, indicating that ambient lighting is more critical than ambient magnetism in sustaining basal cell growth.

The growth and magnetic induction of myoblasts maintained in transparent (“clear”) or opaque (“dark”) chambers were compared to these responses for cells grown under (“open”) ambient lighting to ascertain any environmental interferences produced by the enclosures. Myoblasts grown outside in ambient lighting or within the transparent containers exhibited statistically indistinguishable levels of PEMF-induced proliferation enhancements, indicating that the containment environment per se did not detectably change the culturing conditions (Figure 5A). On the other hand, myoblasts grown under similar conditions of containment, but in the absence of light (dark), exhibited much smaller PEMF responses. Once again, CRY2 overexpression (CRY2) increased basal proliferation (hatched red) and the magnitude of all PEMF responses (hatched blue, green, and black), relative to mock-transfected cells (GFP). These results corroborate that light modulates basal cell proliferation and sensitivity to magnetic fields. TRPC1 and Cyclin D1 protein levels were preferentially upregulated by PEMF exposure of the open, or encased clear, conditions relative to unexposed open or exposed dark conditions, respectively (Figure 5B).

The genetic silencing of the *CRY2* gene further corroborated its contribution to magnetic responses. The negative control (NC) scenario gave similar cell numbers as naïve C2C12 cells, indicating that the transfection protocol was minimally damaging (Figure 6A). CRY2 knockdown (KD; hatched red) produced a 21% decrease in live cell count compared to NC in the basal state (control; solid red). PEMF exposure (solid blue) resulted in 23% and 24% increases in cell number for naïve and NC cells, respectively, relative to their respective unexposed scenarios (solid red). By contrast, PEMF exposure of the CRY2 KD (hatched blue) cells gave only a 16% increase in cell number. CRY2 knockdown thus decreased both the basal proliferative capability as well as reduced the sensitivity of myoblasts to magnetic exposure. Assessment of CRY2 KD protein expression (Figure 6B) revealed a loss of PEMF-induced (i) TRPC1 upregulation and the reduction in cell-cycle-dependent (ii) Cyclin B1 and (iii) Cyclin D1, as well as key signaling proteins such as (iv) phosphorylated ERK and (v) MyoD, corroborating the diminished magnetic sensitivity in these cells.

FAD is an essential co-factor for cryptochrome function [19]. The contribution of FAD to CRY2-dependent magnetoreception was investigated by silencing riboflavin kinase (RFK). In a rate-limiting step in the FAD biosynthetic pathway, RFK phosphorylates riboflavin to form flavin mononucleotide that is then adenylated to form FAD [20]. Knockdown of RFK is known to reduce FAD and CRY2 levels [20]. With reference to responses to magnetic field directionality, RFK NC and naïve myoblasts responded similarly. Downward magnetic fields produced 32% and 34% increases in cell number for naïve and RFK NC cells, respectively, whereas upward magnetic fields gave smaller increases of 18% and 21%, respectively, relative to their respective controls (solid red). On the other hand, in RFK KD cells, upward and downward magnetic fields (hatched blue) resulted in more modest 16% and 25% increases in cell number. At the protein level (Figure 7B), RFK KD cells also exhibited a loss of PEMF-induced (i) TRPC1 expression, similar to CRY2 KD cells with diminished responses in (ii) Cyclin B1, (iii) Cyclin D1, (iv) phosphorylated ERK, and (v) MyoD expressions (Figure 7B). Therefore, RFK knockdown reduced basal proliferative capacity, sensitivity to magnetic fields, and preference for magnetic field directionality.

The previous results showed that TRPC1 expression significantly increased upon PEMF exposure and CRY2 overexpression, whereas TRPC1 expression decreased following CRY2 and RFK silencing, as well as under conditions of reduced ambient light and magnetic isolation. Immunofluorescence microscopy was used to investigate potential physical interactions and colocalization of CRY2 and TRPC1. Two hours following PEMF exposure, both CRY2 and TRPC1 were detected to be evenly distributed between the cytosolic and nuclear compartments (Figure 8). Four hours after PEMF exposure, however, the cytosolic localization of TRPC1 and CRY2 decreased, while nuclear staining for TRPC1 and CRY2 started to predominate. By the sixth hour of PEMF exposure, nearly all of the CRY2 and TRPC1 signals were localized to the nucleus with minimal staining in the cytoplasm, indicating that both proteins had translocated to the nucleus. On the other hand, in the unexposed scenario (Control), no dramatic changes were observed in the distribution of CRY2 and TRPC1 over the entire 6 h interval.

The contribution of light to the nuclear translocation of CRY2 was next investigated. PEMF exposure in the dark (Dark + PEMF) precluded the translocation of CRY2 into the nucleus, whereas the nuclear translocation of CRY2 was enhanced by PEMF exposure under lighted conditions (Light + PEMF), relative to cells not receiving PEMF stimulation under lighted conditions (Controls) (Figure 9A). Quantification revealed a significant increase in CRY2 nuclear localization following PEMF exposure in the presence of light compared to the “Control” and “Dark + PEMF” conditions, whilst no difference between “Control” and “Dark + PEMF” was detected (Figure 9B).

CRY2-overexpression upregulated basal TRPC1 expression that was further augmented by PEMF exposure (Figure 3, Figure 4 and Figure 5) and subsequently resulted in their colocalization to the nucleus (Figure 8). The nature of the interaction between TRPC1 and CRY2 following PEMF exposure was investigated using co-immunoprecipitation. Naïve C2C12 myoblasts were first transfected with a CRY2-FLAG plasmid for purposes of reliable detection. Six hours following PEMF exposure, cytosolic and nuclear fractions were probed for CRY2-specific interaction with TRPC1 (Figure 10). Immunoprecipitation of CRY2 using FLAG antibody revealed a consistent 66 kDA band corresponding to the CRY2 fusion protein both in the input and across all fractionated cytosolic and nuclear fractions. Notably, TRPC1 (~70 kDA) was detected in the immunoprecipitates for CRY2-FLAG across all conditions. The specificity of the CRY2-TRPC1 interaction was confirmed using an IgG control, which showed no signal. These results suggest a direct interaction between CRY2 and TRPC1.

## 4. Discussion

As a category, TRP channels function as biophysical signal transducers [22] of developmental importance [23]. TRP channels accomplish this feat via a process of heteromultimerization that unites the diverse gating sensitivities of different TRP channel subunits into a single tetrameric channel complex [24,25]. The TRPC1 subunit appears to be pivotal for this integrative nature of TRP channels, both within the TRPC family [26] as well as across other TRP families [25]. Accordingly, TRPC1 has been shown to be involved in the transduction of light, mechanical forces, magnetic fields, heat, redox status, pH, and/or calcium levels into cellular responses [22,23].

### 4.1. TRPCs in Photomodulatory Responses

A presumed role for mammalian TRP channels in phototransduction is not without basis [27]. The first TRP channel [28,29] was identified on the basis of a visual transduction mutation in the fruit fly, *Drosophila melanogaster* [30]. This mutant fly was blind in sustained light and exhibited a transient receptor potential (TRP) in response to a flash of bright light due to a defective phototransduction channel within the fly retina. The “canonical” TRPC superfamily are the mammalian homologues of this original *Drosophila* phototransduction channel, with TRPC1 being its founding member [31].

Within the mammalian retina, the TRPC family has been implicated in non-image-forming visual responses mediated by a G-protein-coupled photopigment known as melanopsin [32]. Melanopsin was originally described in intrinsically photosensitive retinal ganglion cells, where it was shown to contribute to circadian photoentrainment and the pupillary light reflex. Melanopsin photoactivation was shown to open TRPC channels, effectively transducing the reception of light information into a downstream response cascade. Melanopsin is now recognized to play signaling roles outside the visual system in tissues not normally reached by external light, including arterial relaxation, regulating lipogenesis, and thermogenesis [33]. An assistive role for biophotons in these hidden tissues remains to be thoroughly examined [34]. Conducting a gene synteny analysis, Bocchero et al. (2020) noted that the TRPC1 gene is closely associated with genes encoding proteins involved in the vertebrate phototransduction cascade and that TRPC1 belongs to the paralogon that comprises the visual G-protein receptor kinases, the arrestins, and the visual guanylyl cyclases [10]. Certain forms of extra-retinal circadian rhythm entrainment may thus be examples of photomodulatory responses recruiting TRPC1 for developmental fruition.

Notably, this study demonstrated that the expression of TRPC1 was upregulated in muscle cells by magnetic exposure and CRY2 overexpression, yet downregulated by magnetic shielding, darkness, CRY2, and FAD silencing that were correlated with parallel changes in markers of myogenic proliferation. These data provide a good basis for initially assigning a role for TRPC1 in cryptochrome-dependent, circadian rhythm-regulated, developmental responses via magnetic and/or phototonic means.

### 4.2. Mammalian Cryptochromes

The cryptochromes (CRYs) are flavoproteins that are best known as blue light phylogenetically related photoreceptors in plants, insects, and animals, including humans [35]. The cryptochromes have also been implicated in diverse forms of magnetoreception, but with overall less extensive evidence to support this role [1]. The first CRY proteins were identified in plants (*Arabidopsis thaliana*), where they were found to be involved in phototropic responses (hypocotyl growth) to blue light [36] that correlated with calcium entry [37,38]. Two mammalian CRYs exist, CRY1 and CRY2 [39,40]. The presence of CRY2 in the pineal gland and in the retina (photoreceptors and ganglion cells) and its light-regulated translocalization to the cell nucleus alludes to a role as a clock protein involved in circadian rhythm entrainment [41]. CRY2 also contributes to photomechanical transduction in skeletal muscle [3,19] as well as in regulating myogenic proliferation [4]. Furthermore, ocular CRY2 has also been shown to underlie directional magnetoreception in cockroaches [42].

### 4.3. Radical Pair Mechanism of Magnetoreception

The cryptochromes are evolutionarily related to a class of DNA repair enzymes known as the photolyases that, upon the photoactivation of an associated flavin cofactor, repair ultraviolet radiation-induced pyrimidine dimers aberrantly formed between DNA strands [1,19,35]. The cryptochromes, whilst having lost their ability to repair damaged DNA, have retained the conserved photolyase homology region (PHR) as well as have acquired unique and isoform-specific C-terminal extensions (CCT) that confer upon the protein signal transduction capabilities specific for each species of cryptochrome. The PHR domain hosts an FAD cofactor that, upon its photoreduction, produces a conformational change in the CCT that exposes it for interaction with downstream signal transducing molecular elements. In the context of the current study, it is possible that CRY2 interacts with TRPC1 via its unique CCT domain upon photoactivation.

The cryptochrome FAD-binding pocket lies within its conserved PHR domain. Upon isolation, the mammalian cryptochromes, CRY1 and CRY2, are found associated with subsaturating levels of FAD. Whereas FAD supplementation stabilizes the expression levels of endogenous and transfected CRY1 and CRY2, preventing FAD biosynthesis by silencing riboflavin kinase expression decreased the endogenous and transfected levels of these cryptochromes [17,20]. Changes in the FAD occupancy of CRY PHRs are thus dynamic and dictate protein stability and enzymatic function. Thus, it is noteworthy that the downregulation of FAD and CRY2 both reduced TRPC1 expression, suggesting some level of mechanistic interaction. The PHR is also the site for the docking of the ubiquitylation ligases that compete with FAD for occupancy of the PHR and thus determine cryptochrome lifetime and circadian periodicity.

As a general modus operandi for cryptochromes, light changes the redox status of flavin cofactors within the PHR domain that then alters the binding interactions of the CCT domain with downstream molecular signaling partners. Mechanistically, photoreduction of the FAD cofactor within the PHD results in an allosteric change in the protein structure that exposes the CCT for phosphorylation by kinases that increases its molecular mass, seen as an upward shift in Western blots (Figure 10). In terms of circadian regulation, following photoactivation, CRY1 and CRY2 associate with the period proteins (PER1 and PER2) that, on the one hand, block the binding of ubiquitylation ligases to the PHR domain to prevent their cytosolic degradation that, on the other hand, permits the translocation of the CRY-PER complex into the nucleus where they, in turn, inhibit their own genetic transcription [17]. CRY-CRY transcriptional repression persists for one diurnal cycle that is determined by the lifetime of the complex (time-dependent degradation) and thus, the commencement of a new photoactivation cycle. Nonetheless, evidence exists both in favor [3] and against [19] photoactivation of CRY1 and CRY2 prior to their induction of biological responses. Photoactivation of CRY2, as supported by the present study, may thus represent a fundamental basis for light-entrained circadian cycling and tissue development [4,41].

Cryptochromes have been convincingly implicated in a form of magnetoreception known as the Radical Pair Mechanism, or RPM. The RPM dictates that weak magnetic fields of a specified cross-orientation and field characteristics should modulate spin-dependent chemical reactions that lie upstream of signal transduction cascades [43]. At the heart of the classical RPM is a light-induced radical pair formed between the cryptochrome host and its FAD factor. In the RPM, light facilitates the ability of magnetic fields to alter the spin state of the radical pairs to change the balance of activated products (forward reaction) to resting state reactants (backward reaction) [14]. Fully photoreduced FAD (flavin semiquinone) has also been proposed to form radical pairs with superoxide radicals in a spin-correlated reaction that serves as a branch point for ROS production, or neutralization, in a magnetically tunable manner [44]. Overall, our results are consistent with both RPM models, either at the level of ROS modulation by photoreduced FAD [9] or as a source of Ca^2+^ with which to regulate mitochondrial function [9,12,45] via the here suggested modulation of TRPC1 by CRY2 (Figure 8 and Figure 10).

In accordance with the above prediction, the same PEMF protocol employed in the present study was previously demonstrated to be capable of generating a mitohormetic level of ROS that could be abolished by pharmacologically blocking TRPC1-mediated Ca^2+^ fluxes [9,12]. Exposure to 1.5 mT PEMFs produced a low level of ROS [9] whose absolute levels and adaptive attributes were increased by downward PEMF exposure [12], indicating that myogenic proliferation is a mitohormetic process. These results agree with the published findings that radical pair dynamics are invoked by orthogonal external magnetic fields that induce mitochondrial respiration and ROS formation to stimulate cellular proliferation [44,46]. Notable differences in the magnetic field paradigms used by Usselman et al. [44,46] and the present study are the signal characteristics and the nature of the field anisotropy. In the former, the amplitude and frequency of the time-variant magnetic field component were set to the Zeeman resonance (1.4 MHz at 50 µT), and the anisotropic static component (50 µT) was experimentally assigned. In the present study, the amplitude of the time-variant magnetic component was 1500 µT, and its orientation was adjusted relative to the ambient static component. These differences are not prohibited in both scenarios invoking an RPM-based mechanism of magnetotransduction [43].

Figure 7 shows that a downward-directed field exposure yielded a higher cell proliferation as compared to an upward-directed field exposure in myoblasts, whereas directionality specificity was abrogated following RFK knockdown that would reduce cellular FAD content, a needed cofactor for cryptochrome function, and consequently, an RPM response. Circadian rhythm entrainment regulates cellular proliferation and differentiation [47]. Accordingly, CRY2 has been shown to be crucial for myogenic cell cycle progression and downstream differentiation [3]. Circadian rhythm modulation of cellular differentiation is mediated by CRY2 and Bclaf1, which stabilize the mRNAs encoding for Cyclin D1 and Tmem176b, regulators for the G1/S phase transition and myogenic cell fusion, respectively [4]. Store-operated calcium entry (SOCE), a process wherein TRPC1 has been implicated [24], also plays a role in photomechanical transduction by CRY2 upstream of myogenic development [3]. This is consistent with the data presented in this study, whereby CRY2-overexpressing cells exhibited elevated Cyclin D1 levels, whereas CRY2 silencing downregulated Cyclin D1 levels [4]. Cyclin D1 levels had been previously shown to be positively correlated with TRPC1 expression induced by magnetic field exposure alone [9] or in combination with light [14]. Complementary data thus exist across diverse studies indicating that TRPC1 and CRY2 are recruited to promote cell cycle progression in response to magnetic exposure.

## 5. Conclusions

This study provides a framework for a developmental interaction between magnetism and cryptochrome-mediated photosensitive circadian responses that have long been presumed, yet unsuccessfully sought. The evidence provided in the present study suggests that magnetosensitivity is conferred upon circadian entrainment via an interaction of TRPC1 with CRY2 that is synergistically induced by both light and magnetic fields. In this study, ambient lighting was used to elucidate cryptochrome contribution to provide relevance to the typical laboratory and clinical environments. Future studies will examine this system in response to the definition of wavelengths of light previously shown to modulate cryptochrome function in the context of the radical pair mechanism. This study also suggests that magnetic field directionality will also need to be taken into consideration when designing PEMF paradigms for developmental efficacy.

## Figures and Tables

**Figure 1 cells-14-00231-f001:**
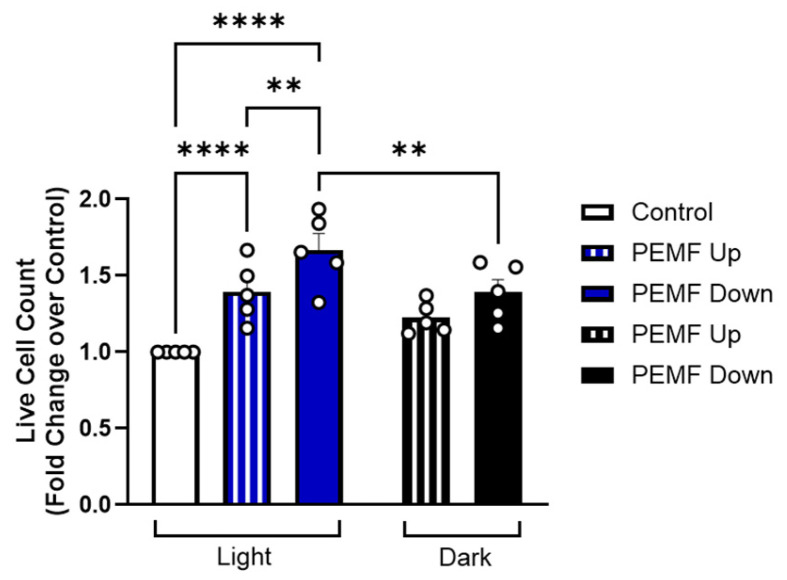
Myogenic proliferation in response to upward- and downward-directed PEMF in light and dark conditions. Live cell count of murine C2C12 myoblasts in response to upward- (lined) and downward-directed (solid) PEMF exposure in both light (in the open) (blue) and dark (black box) (black) conditions compared to light control. Statistical analyses were performed minimally in three independent biological replicates. Data represent n = 5 (with three technical replicates each) independent biological replicates, and statistical analysis was performed using one-way ANOVA with multiple comparison tests, with ** *p* < 0.01 and **** *p* < 0.0001. Error bars represent the standard error of the mean (SEM).

**Figure 2 cells-14-00231-f002:**
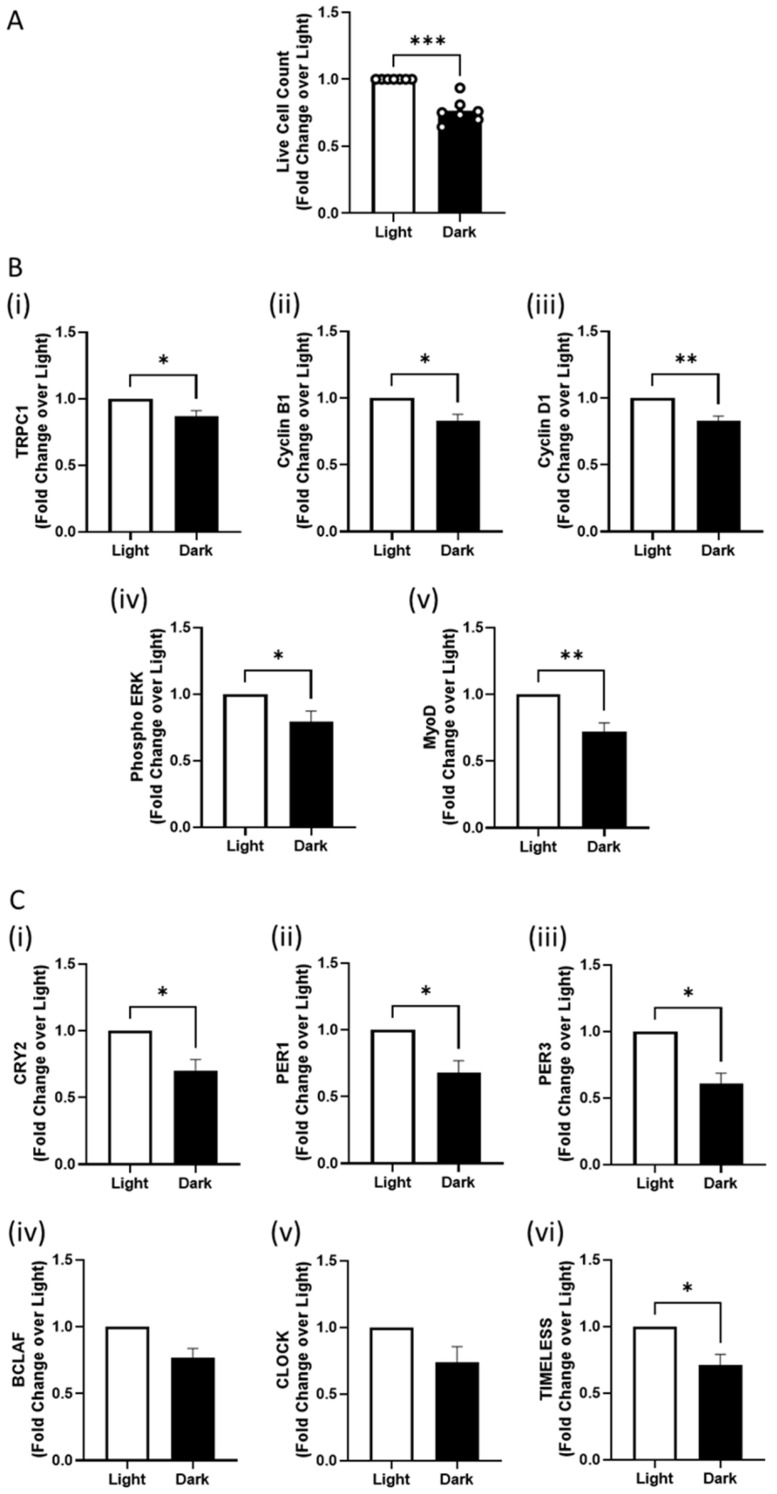
Decrease in cell proliferation and circadian rhythm markers by dark. (**A**) Live cell count of murine C2C12 myoblasts in response to exposure to light (white) and non-exposure/dark (black). (**B**) Protein expression of (i) TRPC1, (ii) Cyclin B1, (iii) Cyclin D1, (iv) phosphorylated ERK, and (v) MyoD in response to light exposure. (**C**) qPCR analysis of basal C2C12 cells in light against dark conditions for 48 h with reference to circadian rhythm regulator markers (i) *CRY2*, (ii) *PER1*, (iii) *PER3*, (iv) *BCLAF*, (v) *CLOCK,* and (vi) *TIMELESS*. Data represent n = 7 (with three technical replicates each) independent biological replicates, and statistical analysis was performed using a standard t-test with * *p* < 0.05, ** *p* < 0.01, and *** *p* < 0.001. Error bars represent the standard error of the mean (SEM).

**Figure 3 cells-14-00231-f003:**
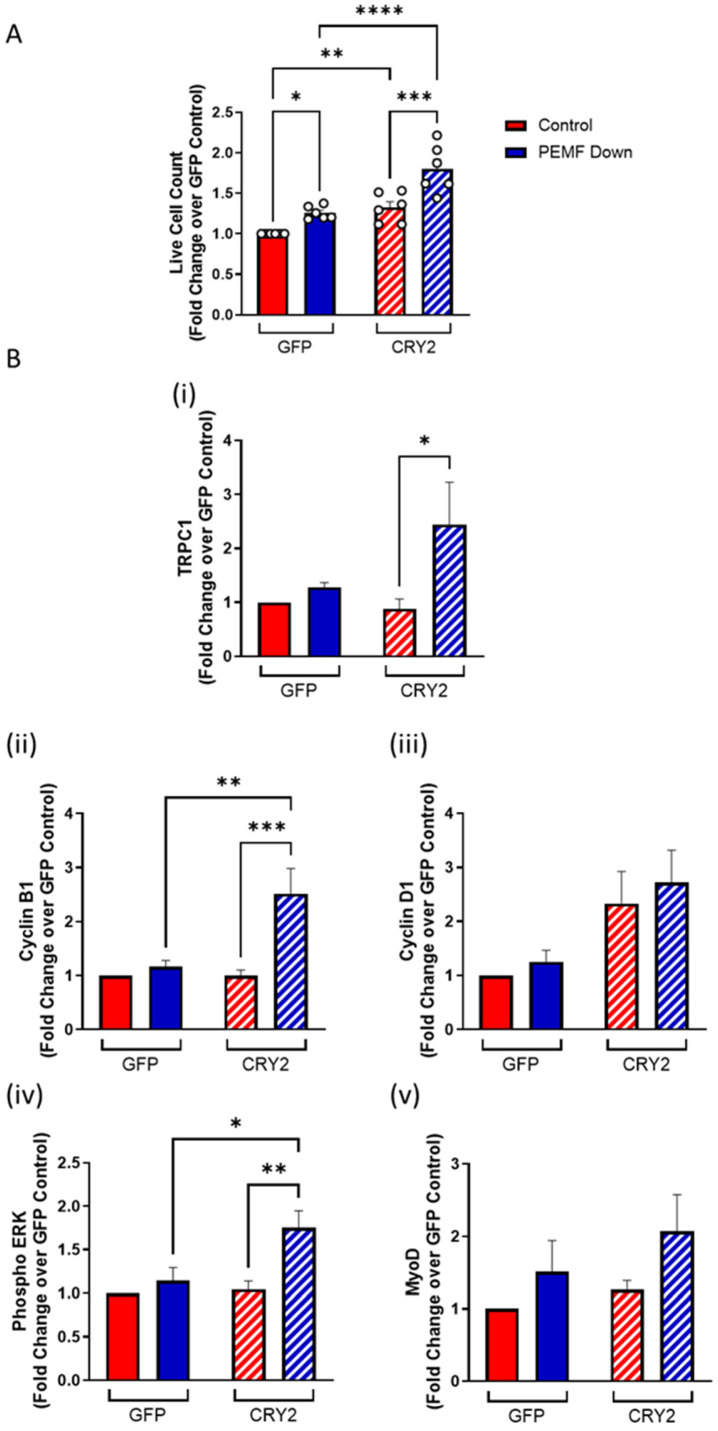
Increase in cell proliferation with PEMF exposure in vector and CRY2-overexpressing cells. (**A**) Live cell count of C2C12-GFP vector (solid) and C2C12-CRY2-overexpressed (hatched) myoblasts in response to PEMF exposure (blue) compared to 0 mT control (red). (**B**) Protein expression of (i) TRPC1, (ii) Cyclin B1, (iii) Cyclin D1, (iv) phosphorylated ERK, and (v) MyoD in response to PEMF exposure. Cell enumerations and protein collection were conducted 48 h after plating and 24 h after PEMF or sham exposure. Data represent n = 6 (with three technical replicates each) independent biological replicates, and statistical analysis was performed using one-way ANOVA followed by multiple comparison tests with * *p* < 0.05, ** *p* < 0.01, *** *p* < 0.001, and **** *p* <0.0001. Error bars represent the standard error of the mean (SEM).

**Figure 4 cells-14-00231-f004:**
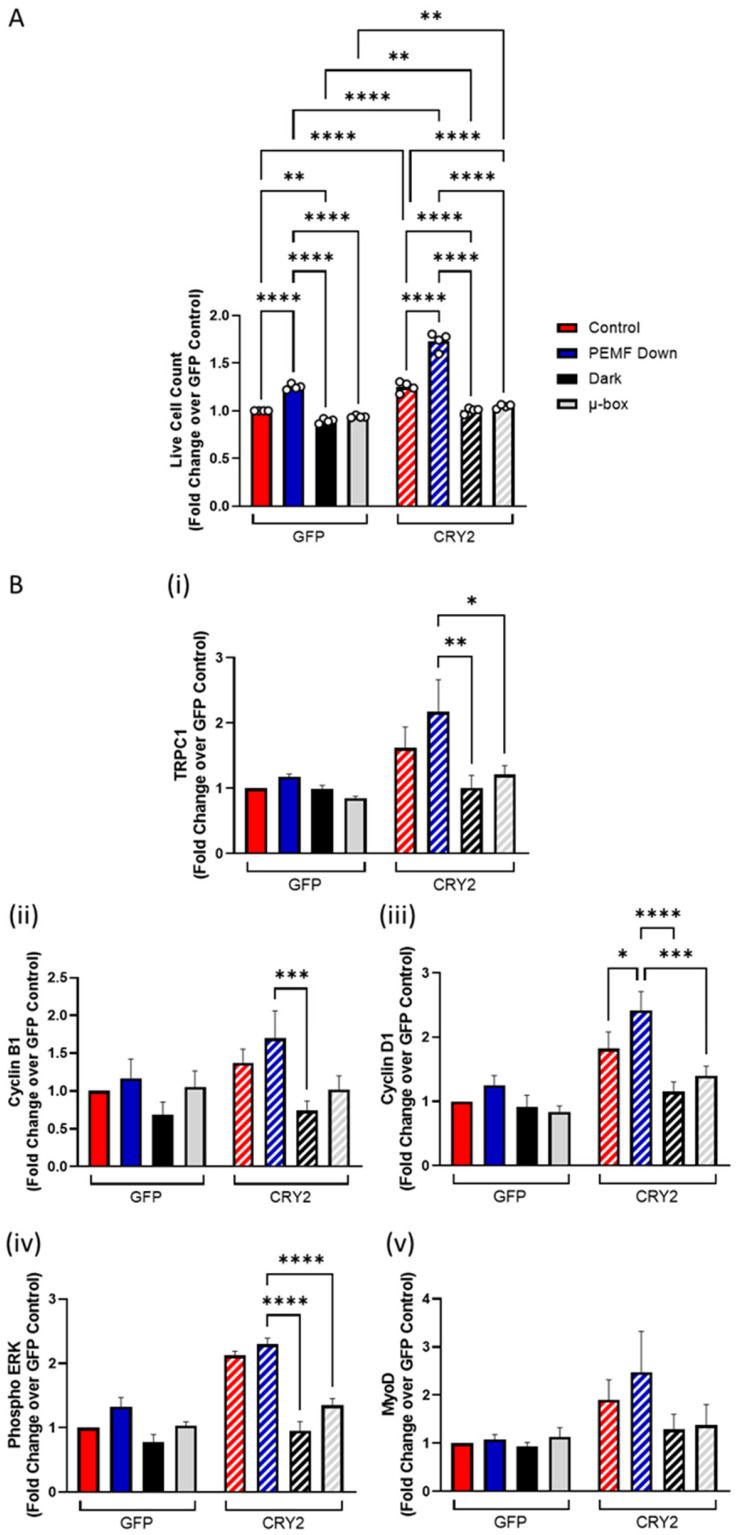
Myogenic proliferation in response to PEMF exposure, dark and shielding from ambient magnetic fields. (**A**) Live cell count of GFP vector cells and CRY2-overexpressing cells in response to different stimuli of control/light using the transparent box (red), PEMF exposure (blue), dark (black), and µ-box (gray) as a fold change over GFP control. Cell enumerations and protein collection were conducted 48 h after plating and 24 h after PEMF or sham exposure. (**B**) Protein expression of (i) TRPC1, (ii) Cyclin B1, (iii) Cyclin D1, (iv) phosphorylated ERK, and (v) MyoD in response to control/light, PEMF exposure, dark, and ambient magnetic field shielding conditions for both GFP vector (solid) and CRY2-overexpressed (hatched) myoblasts. Data represent n = 4 (with three technical replicates each) independent biological replicates, and statistical analysis was performed using one-way ANOVA followed by multiple comparison tests with * *p* < 0.05, ** *p* < 0.01, *** *p* < 0.001, and **** *p* < 0.0001. Error bars represent the standard error of the mean (SEM).

**Figure 5 cells-14-00231-f005:**
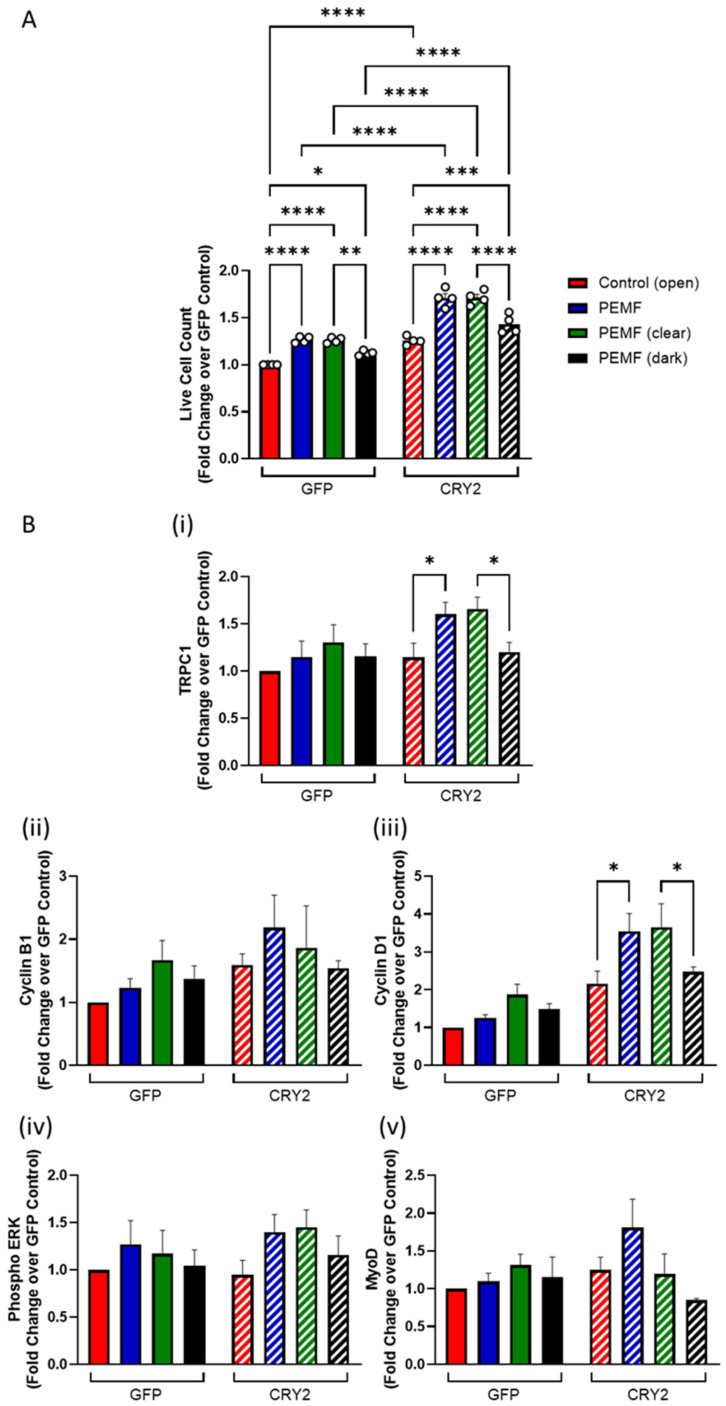
Comparison of myogenic potential between normal PEMF exposure and PEMF exposure using the transparent and dark boxes. (**A**) Live cell count of GFP vector cells and CRY2-overexpressing cells in response to different stimuli of control/light in the open (red), regular PEMF exposure without any encompassing box (blue), PEMF exposure using the clear box (green), and PEMF exposure in the dark with the dark box (black) as a fold change over GFP control. Previous experiments with PEMF exposures did not utilize the clear box as the clear box was used for the control condition. This experiment utilized the clear box as a PEMF condition to compare how having an encompassing box would affect cellular responses to regular PEMF exposure without a box. (**B**) Protein expression of (i) TRPC1, (ii) Cyclin B1, (iii) Cyclin D1, (iv) phosphorylated ERK, and (v) MyoD in response to control/light, PEMF exposure without any encompassing box, PEMF exposure using a clear box, and PEMF exposure in the dark for both GFP vector (solid) and CRY2-overexpressed (hatched) myoblasts. Data represent n = 4 (with three technical replicates each) independent biological replicates, and statistical analysis was performed using one-way ANOVA followed by multiple comparison tests with * *p* < 0.05, ** *p* < 0.01, *** *p* < 0.001, and **** *p* < 0.0001. Error bars represent the standard error of the mean (SEM).

**Figure 6 cells-14-00231-f006:**
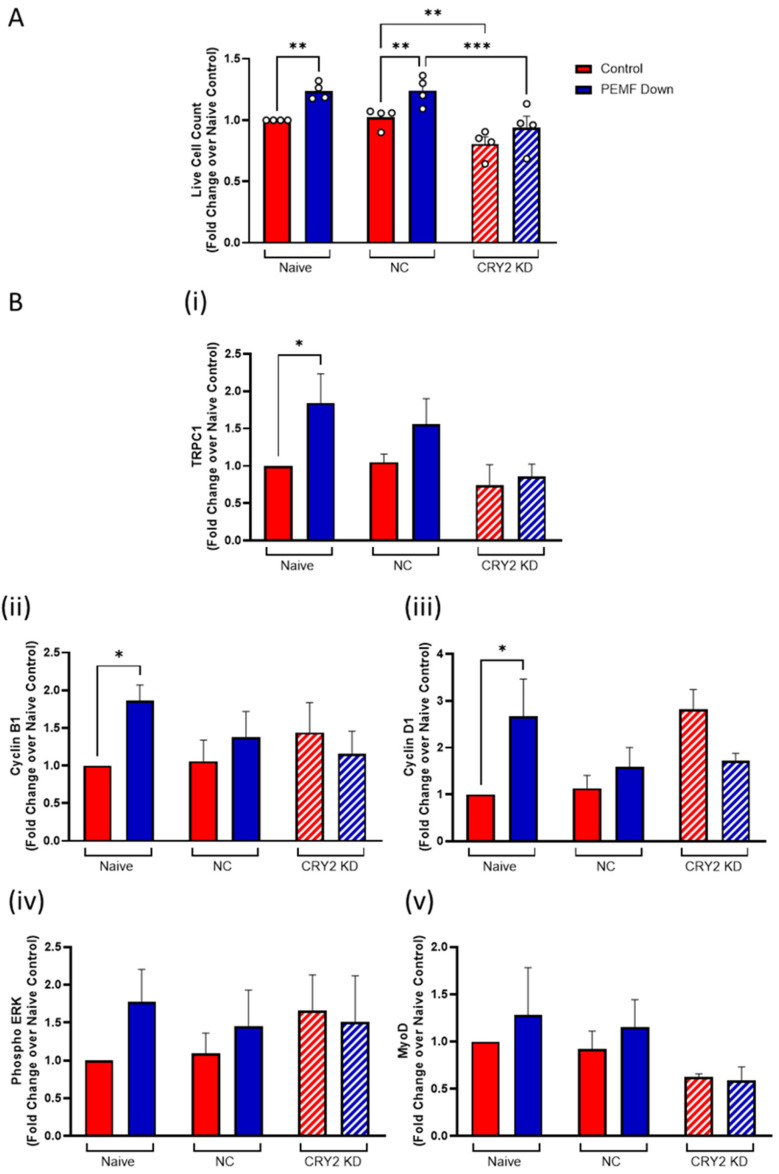
Effect of CRY2 silencing on myogenic proliferation. (**A**) Live cell count of CRY2 knockdown (KD) C2C12 cells, scrambled dsiRNA C2C12 cells (NC), and naïve C2C12 cells following PEMF exposure. (**B**) Protein expression of (i) TRPC1, (ii) Cyclin B1, (iii) Cyclin D1, (iv) phosphorylated ERK, and (v) MyoD in response to PEMF exposure. C2C12 murine myoblasts were transfected with CRY2 KD dsiRNA or scrambled dsiRNA 24 h post-seeding, and PEMF treatment was given 24 h after. Solid red bars denote cell number with no PEMF treatment as controls, while solid blue bars denote that with PEMF treatment at 1.5 mT for naïve and NC cells, whereas the respective bars for CRY2 KD cells were hatched. Data represent n = 4 (with three technical replicates each) independent biological replicates, and statistical analysis was performed using one-way ANOVA followed by multiple comparison tests. Significance levels are indicated as follows: * *p* < 0.05, ** *p* < 0.01, and *** *p* < 0.001. Error bars represent the standard error of the mean (SEM).

**Figure 7 cells-14-00231-f007:**
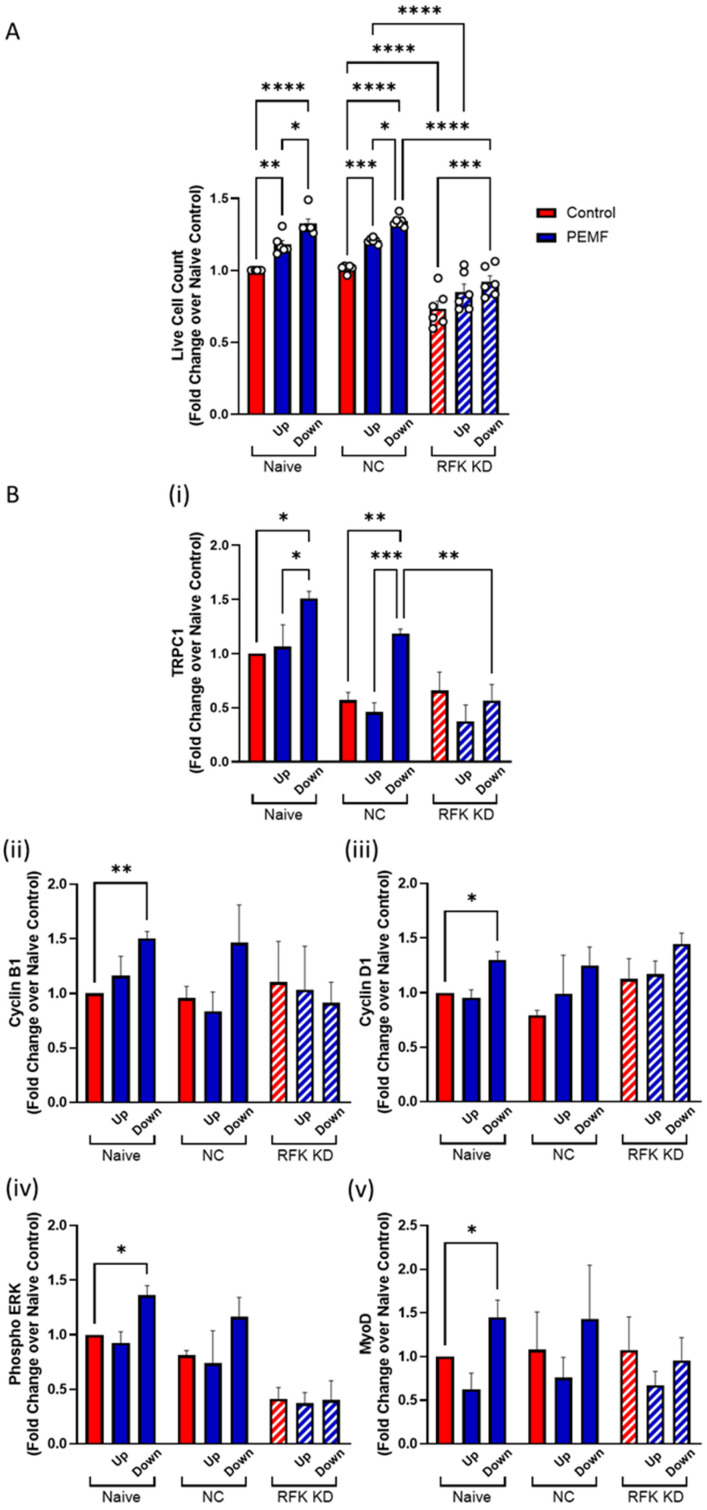
Effect of CRY2 precursor knockdown of RFK on myogenic proliferation and abolishment of directionality–specificity. (**A**) Cell numeration of RFK knockdown (KD) C2C12 cells and scrambled dsiRNA C2C12 cells against naïve C2C12 cells after PEMF treatment for 24 h. (**B**) Protein expression of (i) TRPC1, (ii) Cyclin B1, (iii) Cyclin D1, (iv) phosphorylated ERK, and (v) MyoD in response to PEMF exposure. C2C12 murine myoblasts were transfected with RFK KD dsiRNA and scrambled dsiRNA 24 h post-seeding, and PEMF treatment was given 24 h after. Red bars denote cell number with no PEMF treatment as controls, while blue bars denote that with PEMF treatment with upwards and downwards field directionality at 1.5 mT, where solid bars denote naïve and NC cells while hatched bars represent RFK KD cells. Data represent n = 6 (with three technical replicates each) independent biological replicates, and statistical analysis was performed using one-way ANOVA followed by multiple comparison tests with * *p* < 0.05, ** *p* < 0.01, *** *p* < 0.001, and **** *p* < 0.0001. Error bars represent the standard error of the mean (SEM).

**Figure 8 cells-14-00231-f008:**
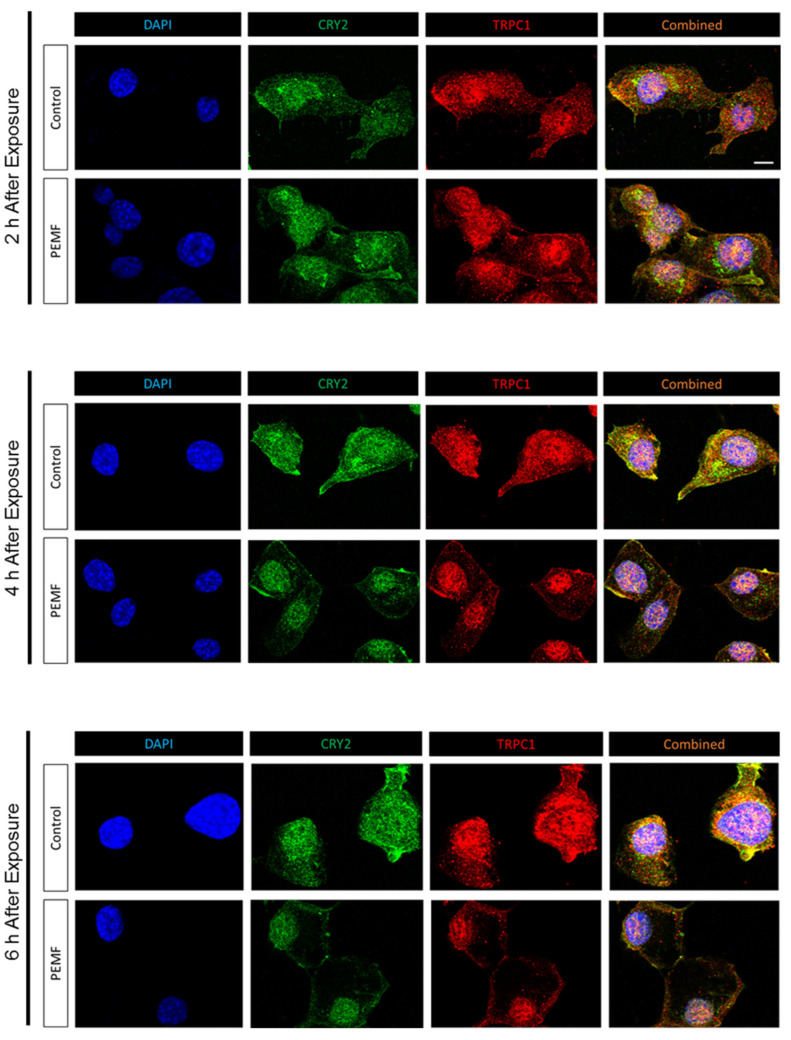
Time-course analysis of CRY2 and TRPC1 protein translocation into the nucleus. Representative images of naive C2C12 myoblasts stained with CRY2 and TRPC1 at 2, 4, and 6 h post PEMF exposure. Scale bar = 10 μm.

**Figure 9 cells-14-00231-f009:**
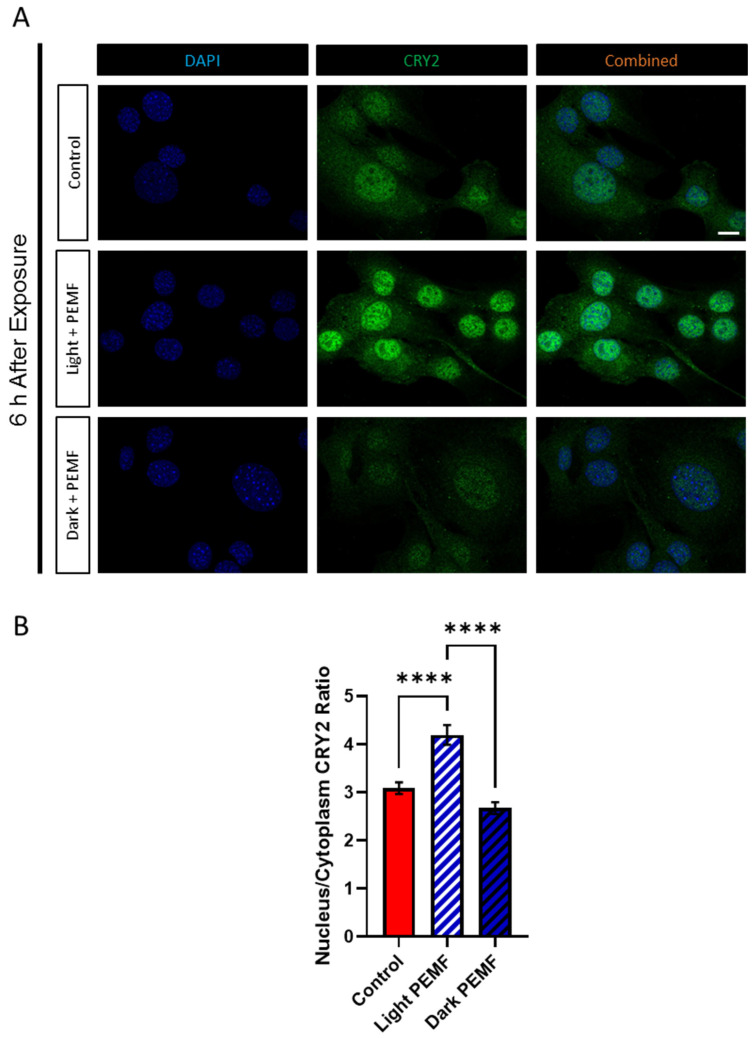
CRY2 protein localization 6 h post PEMF exposure in the presence or absence of light. (**A**) Representative images of naive C2C12 myoblasts stained with CRY2 under different conditions: control (light with no PEMF), PEMF exposure in the light (clear box), and PEMF exposure in the dark (black box). Scale bar = 10 μm. (**B**) Quantification of CRY2 translocation into the nucleus, calculated as the nuclear/cytoplasmic ratio. Data represent three independent biological replicates, and statistical analysis was performed using one-way ANOVA followed by multiple comparison tests with **** *p* < 0.0001. Error bars represent the standard error of the mean (SEM).

**Figure 10 cells-14-00231-f010:**
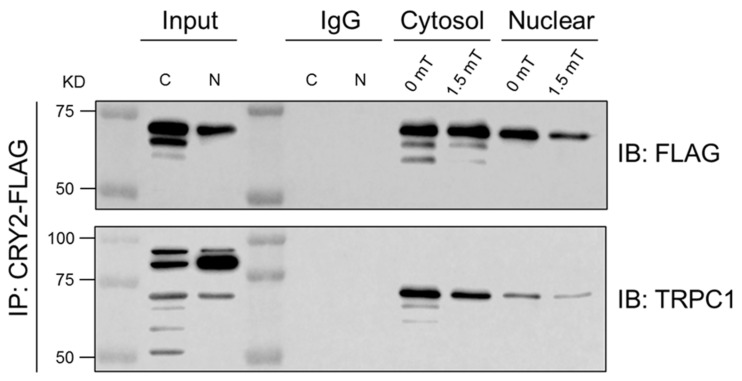
TRPC1 interaction with CRY2 via immunoprecipitation. CRY2-FLAG transfected C2C12 cells were immunoprecipitated with FLAG antibody and analyzed for TRPC1 using Western blot analysis. CRY2-FLAG protein was detected at 66 kDa and TRPC1 at ~70 kDa [21] in input, cytosolic, and nuclear fractions, but not in the IgG (immunoglobulin) control (n = 2).

**Table 1 cells-14-00231-t001:** List of primary antibodies used for Western blot analysis.

Antibody Name	Dilution Factor	Cat. No.	Manufacturer
GFP	1:300	sc-9996	Santa Cruz
TRPC1	1:300	sc-133076	Santa Cruz
Cyclin B1	1:1000	4138S	Cell Signaling Technology
Cyclin D1	1:300	sc-8396	Santa Cruz
MyoD	1:300	sc-71629	Santa Cruz
Phospho-ERK	1:300	sc-7383	Santa Cruz
ERK	1:300	sc-514302	Santa Cruz
GAPDH	1:10,000	60004-1-1g	Proteintech
FLAG Tag	1:1000	20543-1-AP	Proteintech

**Table 2 cells-14-00231-t002:** List of primers used for qPCR.

Gene	Forward (from 5′ to 3′)	Reverse (from 5′ to 3′)
*CRY2*	GTGCACTGGTTCCGCAAAG	CACGGGTCGAGGATGTAGAC
*PER1*	GAAACGGCAAGCGGATGGAG	GGAGGACGAAACAGGGAAGG
*PER3*	ATTTAAGCACGTGGGGCTCA	GACCCTGCTTGAACACAGGA
*BCLAF*	ATCACATTCTTCAAGATCCAAGTC	CTGGAACGAGACCTAGAACTGTAT
*CLOCK*	TCTCAGACCCTTCCTCCACA	GAGCTGGCAAATGCTGTCTG
*TIMELESS*	CTTCAGAGGACTCGCGGAGA	GGCTCCTTGTGGTAAGTCCC
*RFK*	CAGGTGGTGCGTGGCTTC	CATCCAATGCTCACCACCAT

## Data Availability

All data supporting the results are presented in the manuscript. Any other inquiries can be directed to the corresponding authors via email.

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
