# Peer review of "Magnetically Stimulated Myogenesis Recruits a CRY2-TRPC1 Photosensitive Signaling Axis"

_cells, 2025, doi:10.3390/cells14030231_

Round 1
Reviewer 1 Report
Comments and Suggestions for Authors
I appreciate the opportunity to evaluate this manuscript.
The impact of electromagnetic and magnetic fields on organisms has been the subject of research by many authors for many years. This is a very interesting and important issue, because the number of sources of fields in the environment increases every year. A field with appropriately selected parameters can also serve as a factor used in diagnostics or rehabilitation.
The introduction of reviewed paper is written very well, providing literature data regarding all the relevant subject. The methodology is well-written, thoroughly detailing all essential information on the materials and methods used. The results section is clearly and effectively presented, showcasing the findings in a well-organized manner. The data is conveyed with clarity, making the outcomes easy to understand and interpret. The discussion section is well written, providing insightful interpretations of the results in a clear and coherent manner. It effectively addresses the study's implications, showing a strong understanding of the research context.
I believe that the paper is well written and can be published in its current form. I will await with interest the further research that will be testing this system in response to defined wavelengths.
Reviewer 2 Report
Comments and Suggestions for Authors
Please specify also within Results and Discussion the wavelengths of what you state in Conclusions as "ambient light". Because this is so important also in the night-day entrainment that these are shorter wavelengths and not light bulbs with more red light.
You write in Abstract CRY2 - this may be okay in the Title but please write the complete notation in Abstract.
Reviewer 3 Report
Comments and Suggestions for Authors
General comment: The authors investigated the role of cryptochromes (CRY2), flavoproteins and TRPC1 channels in the response of myoblasts to pulsed electromagnetic fields (PEMFs) and their sensitivity to the directionality of magnetic fields. Growing in the dark reduced the sensitivity of myoblasts to PEMFs, and overexpression of CRY2 enhanced the responses, while depletion of flavins (FAD) abolished this sensitivity. CRY2 and TRPC1 were observed to interact and translocate into the nucleus after exposure to PEMFs, suggesting a CRY2-dependent signaling cascade. The results highlight a magnetotransduction mechanism involving CRY2, with important influences of light and flavoproteins.
Strong points
1. Overall, the article may be of interest to the scientific community. The manuscript is relevant to the field and is well-designed.
2. The materials and methods described in the paper are detailed and well structured.
Minor corrections:
1. I suggest detailed explanations regarding the choice of certain experimental thresholds, such as the duration of PEMF exposure (10 minutes) or the concentrations used.
2. I recommend clearly specifying the number of biological replicates in the materials and methods section for each experimental set. For example: “The experiments were performed in biological and technical triplicates for each condition.”
3. Specify whether the data were tested for normality before statistical analysis (e.g., Shapiro-Wilk test) and mention adjustments made for outliers.
4. Include details about the duration of maintenance under selection (e.g., “Cells were maintained in selective medium for X weeks before use”).
5. “Drosophila” – should be written in italics.
